# Development and Pilot Use of a Questionnaire to Assess the Knowledge of Midwives and Pediatric Nurses on Maternal Use of Analgesics during Lactation

**DOI:** 10.3390/ijerph182111555

**Published:** 2021-11-03

**Authors:** Ine Janssens, Margot Van Hauwe, Michael Ceulemans, Karel Allegaert

**Affiliations:** 1Department of Public Health and Primary Care, Academic Centre for Nursing and Midwifery, 3000 Leuven, Belgium; Ine.janssens@student.kuleuven.be (I.J.); vanhauwemargot@hotmail.com (M.V.H.); 2Clinical Pharmacology and Pharmacotherapy, Department of Pharmaceutical and Pharmacological Sciences, KU Leuven, 3000 Leuven, Belgium; michael.ceulemans@kuleuven.be; 3Woman and Child, Department of Development and Regeneration, KU Leuven, 3000 Leuven, Belgium; 4Department of Clinical Pharmacy, Erasmus MC Rotterdam, Wytemaweg Hospital Pharmacy, 3075 CE Rotterdam, The Netherlands

**Keywords:** questionnaire development, lactation, breastfeeding, analgesics, education, knowledge assessment, midwife, nurse

## Abstract

There is a need to assess the knowledge of healthcare providers on the use of maternal analgesics during lactation; however, valid instruments are not yet available. This study aimed to develop and test a valid questionnaire on the knowledge of analgesics (acetaminophen, ibuprofen, aspirin, tramadol, codeine, oxycodone) during lactation, using a structured, stepwise approach. As a first step, literature was screened to generate a preliminary version consisting of a pool of item subgroups. This preliminary version was subsequently reviewed during two focus groups (midwives: *n* = 4; pediatric nurses: *n* = 6), followed by a two-round online Delphi with experts (*n* = 7) to confirm item and scale content validity. This resulted in an instrument consisting of 33 questions and 5 specific clinical case descriptions for both disciplines. Based on the assumption of an a priori difference in knowledge between midwives and pediatric nurses related to their curricula (known-groups validity), high construct validity was demonstrated in a pilot survey (midwives: *n* = 86; pediatric nurses: *n* = 73). We therefore conclude that a valid instrument to assess knowledge on lactation-related exposure to analgesics was generated, which could be further validated and used for research and educational purposes. As these pilot findings suggest suboptimal knowledge for both professions on this topic, adaptations to their curricula and postgraduate training might be warranted.

## 1. Introduction

Breastfeeding has multiple benefits for both mother (such as lower postpartum blood losses; faster postpartum weight normalization; lower risk for type 2 diabetes, breast or ovarian cancer; lower incidence of osteoporosis) and infant (such as lower risk of gastro-intestinal, respiratory and urinary tract infections; lower risk for type 2 diabetes or obesity; better neurodevelopment) [1]. The use of medicines during lactation is very common, as taking medicines can be appropriate or necessary to protect, improve or restore maternal health. Unfortunately, this can also result in unintended exposure to the nursing infant [2,3]. Therefore, the ultimate goals of maternal medicine use during breastfeeding are dual. First, effective and safe medicines should be provided for a diversity of maternal indications (e.g., postpartum maternal analgesia, maternal co-morbidities, pregnancy/breastfeeding related diseases or vaccinations). Simultaneously, one also aims to ensure the safety of the nursing infant, avoiding both relevant adverse reactions related to unintended exposure as well as unneeded interruption or termination of lactation [2,4,5].

Lactating women are still often advised to discontinue or even stop nursing while taking medicines, although there are only a limited number of medicines that are identified as potentially or likely harmful to the breastfed infant [2,4,5]. Using a prospective study design Ito et al. documented that in a cohort of 838 nursing infants with mothers taking medicines, the incidence of adverse reactions (so claimed to be potentially causally related) in the infants was 11.2% (94/838) [5]. All were classified as minor reactions, not necessitating medical attention. Antibiotics (19.3%), antihistamines (9.4%), sedatives, antidepressants and anti-epileptics (7.1%), but also analgesics and narcotics (11.2%) were most commonly associated with adverse reactions. This list hereby also likely reflects the more commonly used medicines in this population [2,4,5]. A more relevant point is that these data suggest that breastfeeding rarely needs to be discouraged or discontinued when a mother needs pharmacotherapy, but some cautiousness about, for example, analgesics may be warranted [6,7]. Consequently, guidance and guidelines on anesthesia and sedation or on the use of analgosedatives in breastfeeding women have been published, although they are rather focused on the hospital setting [8,9].

Worldwide, a relevant number of the marketed analgesics can be purchased or dispensed without prescription (over-the-counter, OTC). This includes acetaminophen (paracetamol), aspirin and non-steroidal anti-inflammatory drugs (NSAIDs). Currently, acetaminophen is considered to be the most commonly used medicine in the pregnant and breastfeeding population [10,11,12,13]. Midwives and pediatric nurses come across breastfeeding mothers on a daily base and play a crucial role in educating and informing them on the use of analgesics during lactation [11]. The level of knowledge and the potential differences in the level of knowledge between midwives and pediatric nurses on this topic are generally unknown, including for Belgium, but is anticipated to be different based on their specific curricula. Insight into the current knowledge of healthcare providers (HCPs) is, however, highly relevant, as lack of knowledge, and as such giving incorrect, insufficient, conflicting or unclear information, may result in unnecessary interruption or even termination of lactation. Along the same line, this may also lead to inadequate maternal pain management or in adverse events in the nursing infant [14,15,16].

Consequently, there is a need to assess the knowledge and possible deficits on analgesics during lactation among HCPs. However, no existing validated knowledge instruments are available in the literature to provide these insights. This study therefore aimed to develop and pilot test a valid (the extent to which an instrument measures what it claims to measure) instrument to evaluate the knowledge of midwives and pediatric nurses about the use of analgesics during lactation [17]. This knowledge instrument focuses on the following analgesics: acetaminophen, ibuprofen, aspirin, tramadol, codeine and oxycodone.

## 2. Materials and Methods

The study consisted of three parts: (1) development of an instrument to assess the knowledge of midwives and pediatric nurses on the use of analgesics during lactation (i.e., step 1 and 2); (2) content validation of this instrument (i.e., step 3); (3) pilot use of this instrument in a cohort of midwives and pediatric nurses (i.e., step 4). The study obtained ethical approval from the competent ethics committees involved (University Hospitals Leuven; MP016395; 25 February 2021 and GasthuisZusters Antwerpen (GZA); 210202MASTER; 9 February 2021). Participants provided informed consent prior to study contribution.

### 2.1. Development of the Instrument

The knowledge instrument was developed based on a literature search (step 1), followed by focus group discussions with both target populations (i.e., midwives and pediatric nurses) (step 2).

To generate a pool of potentially relevant themes and items for a preliminary questionnaire, a literature scanning and assessment was performed. To perform this, PubMed and Limo (i.e., a search engine within KU Leuven to search the entire KU Leuven library’s collection) were assessed by two researchers (I.J. and M.V.H.) with focus on perceived or reported knowledge (gaps) of either mothers or HCPs on the use of analgesics during lactation. In this explorative search, we used Boolean operators ‘AND’ or ‘OR’ to combine the search words as follows: (knowledge or perception or vision) AND (midwives or nurses or caregivers or mothers or parents) AND (medication or medicine or analgesics) AND (risk or harmful effects) AND (breastfeeding). Included papers described aspects or items related to knowledge (deficiencies) or perceptions of mothers or HCPs on the use of analgesics (non-opioids or opioids) during lactation. Exclusion criteria were any other type of medicines or any other indication (such as abstinence syndrome prevention or treatment). The snowball method was applied to the papers retained, and agreement between both researchers was sought. Information from the selected papers provided input for the development of a preliminary questionnaire with different item subgroups, which were covered by multiple choice questions and clinical case descriptions.

This preliminary questionnaire was subsequently reviewed during two focus groups consisting of certified and professionally active midwives or pediatric nurses, respectively. To do so, convenience sampling was used to recruit midwives working at the maternity ward of the University Hospitals Leuven, Belgium, and pediatric nurses employed at the general pediatric ward of the GZA hospital, campus St. Augustinus, Antwerp, Belgium. These colleagues were questioned about their views and perceptions on the item subgroups, the preliminary set of questions and the clinical case descriptions. The language level and comprehensibility for the target groups were also verified by these participants.

### 2.2. Content Validation

To determine content validity (step 3), an expert panel was composed, covering both content (i.e., medicine, pharmacy and nursing sciences) and methodological expertise on questionnaire development. The current version of the questionnaire, which was available after step 2 of the development process, was assessed by experts in step 3 in a two round e-Delphi approach.

In the first round, content validity was first assessed by all individual experts, scoring all items using a Likert scale approach (1 = not relevant; 2 = partially relevant; 3 = relevant; 4 = very relevant). Based on the individual assessments, the item-level content validity index (I-CVI) [(number of experts with a score 3 or 4)/(total number of experts)], with a targeted index > 0.78 was calculated [17]. Items with a I-CVI < 0.78 were removed.

In the second round, re-assessment of the content validity in the retained items was carried out by experts who were selected from the initial group. Selection criteria hereby were the absence of ‘outlier’ assessment in the first round (low or high scores on a set of items compared to the assessment of these items by the other experts, combined with a qualitative assessment of their individual feedback on the items). Based on their assessment, the I-CVI was recalculated with subsequent assessment of the scale content validity index (S-CVI) [(the sum of the individual I-CVI)/(the number of items)] with a targeted index > 0.90 [17].

### 2.3. Pilot Use of the Instrument

Following instrument development and content validation, the questionnaire was pilot tested in a convenient sample of the target population using an anonymous, cross-sectional e-survey via the Qualtrics platform (www.qualtrics.com, accessed on 1 April 2021) (step 4). To do so, the same criteria on sampling were applied as in step 2 (i.e., certified and professionally active midwives and pediatric nurses). Personal social media channels and direct e-mailing were hereby used to attain a relevant number of responses within a two-week study period. Based on the results obtained in the e-survey, construct validity was assessed assuming an a priori difference in knowledge between midwives and pediatric nurses related to their curricula [17].

Construct validity was hereby assessed for the different item subgroups of the instrument by known-groups validity. For the between group analysis (i.e., comparison of responses from midwives and pediatric nurses), unpaired t-tests and Chi-square tests were used for continuous and dichotomous variables, respectively. Only fully completed questionnaires were considered for analysis. Statistical analyses were conducted using SPSS Statistics version 27 (IBM Corp., Armonk, NY, USA).

## 3. Results

The workflow of the instrument development and testing is summarized in Figure 1, and is based on the development (step 1 and 2), content validation (step 3) and pilot use (step 4, construct validation).

### 3.1. Development of the Instrument

#### 3.1.1. Step 1, Literature Screening

Based on the literature screening performed in December 2020, six illustrative papers were selected, reflecting the different item subgroups or items retained in the preliminary questionnaire (see Table 1 for an overview of these six illustrative papers). These papers demonstrate and reflect the knowledge gaps among HCPs and mothers on the potential risks of lactation-related medicine exposure in nursing infants. This includes the use of analgesics and retrieval of reliable sources of information, while self-reporting on knowledge of analgesics by HCPs was also found to be valuable.

Based on the literature screening, five item subgroups were identified (subgroup 1: infant risks related to respiratory depression, sedation or internal bleeding [16,18]; subgroup 2: milk production or volume related effects [16]; subgroup 3: transfer of medicines to human milk and relevance of doses [18,19,20,21]; subgroup 4: sources to retrieve information [20,22]; subgroup 5: personal assessment of the individual level of knowledge on the use of analgesics during lactation [16]. To avoid bias or gambling, we added the option “I don’t know” to all questions. This resulted in a preliminary questionnaire of 56 common questions and 4 and 7 clinical case descriptions, which were specific for midwives and pediatric nurses respectively, to further assess their knowledge.

#### 3.1.2. Step 2, Focus Groups

Due to the COVID-19 restrictions, both focus groups had to be organized online in February 2021. In total, four midwives (a fifth midwife was unable to attend, but provided her comments by mail) and six pediatric nurses participated. The language level and comprehensibility were assessed to be adequate by both focus groups.

During the focus group with midwives, several suggestions concerning the content of the questionnaire were brought up: doses should be added and should reflect clinical practices for pain management (item subgroup 1 and 2), maximal doses should be questioned (item subgroup 3) and HCPs should be added as sources of information, including lactation specialists, gynecologists, pediatricians and neonatologists (item subgroup 4). The instrument was subsequently modified based on these comments. With regard to the clinical case descriptions, midwives requested some rephrasing to avoid potential misunderstanding and asked for clearer dosing descriptions in the cases.

The focus group of pediatric nurses preferred using the word ‘risk’ instead of side effect (as side effects technically could be also positive (item subgroup 1)), while dosing related comments were also provided (item subgroup 2 and 3). For item subgroup 4, ‘a colleague nurse from the unit’ was also suggested as another source of information. With regard to the clinical case descriptions, the pediatric nurses considered the oxycodone case as not relevant for their practice and suggested some additional rephrasing to improve clarity. This included, for example, consistent use of either brand or generic names.

Based on the feedback received from the midwives and nurses, the items in the questionnaire were reduced to 46, compared to the 56 questions after step 1, while 2 × 6 clinical case descriptions were retained, although rephrased to some extent. The five item subgroups were ‘risks (respiratory distress, sedation or bleeding) for the infant during the maternal use of analgesics during lactation’, ‘let-down reflexes’, ‘safety related to short and prolonged use of analgesics during lactation’, ‘access to and use of sources of information’ and ‘self-reported knowledge’.

### 3.2. Content Validation of the Instrument

The expert panel participating in the e-Delphi consisted of seven members (i.e., lactation specialist (*n* = 1), neonatologist (*n* = 1), pharmacists (*n* = 2), pediatrician (*n* = 1) and teachers involved in the bachelor-after-bachelor program in midwifery and pediatric nursing (*n* = 2)) and was organized in March 2021 (step 3). In the first round, the seven experts reviewed the items and clinical case descriptions of relevance and provided comments. For 39 items (questions or clinical case descriptions), the I-CVI score was ≥0.85, 9 items had a score < 0.71, 10 items had a score of 0.71 and were provisionally retained.

After reviewing I-CVI and re-contacting experts in case of disagreement, it was decided to remove 17 questions and one clinical case description per target group (19 items). Furthermore, additional rephrasing of some item subgroups was suggested. The final five item subgroups were ‘risks for the infant during the maternal use of analgesics during lactation (respiratory depression, sedation and bleeding)’, ‘let-down reflexes’, ‘safe short and prolonged use of analgesics during lactation’, ‘access to and use of sources of information’ and ‘self-reported knowledge’. The proportion of relevance per expert was then calculated. Based on the quality of the feedback and proportion relevance, less qualitative experts were not invited for the second round.

The second round included three experts from the initial pool (proportion relevance of 0.86, 0.91 and 0.82 respectively), who were contacted again to score the current version of the instrument for relevance and to add suggestions. Based on their input, a final version of the questionnaire was compiled. The final questionnaire consisted of 33 questions (i.e., 17 knowledge questions, 8 questions on access to and use of sources of information and 8 questions on self-reported knowledge) and five clinical case descriptions per target group. The questionnaire showed excellent content validity because the I-CVI and S-CVI scores were 1 at the end of the expert panel step. In other words, the final instrument with 33 questions and 5 clinical case descriptions per target group was agreed by all experts and subsequently published electronically for the pilot use. The Dutch and English version of the final instrument is provided in the Appendix A, along with the correct answers and scoring instructions for the knowledge questions.

### 3.3. Pilot Use of the Instrument: Construct Validity Based on Known-Groups Validity

Based on 253 survey initiations, 159 HCPs (midwives: *n* = 86; pediatric nurses: *n* = 73) fully completed the questionnaire in April 2021 (i.e., completion rate of 68%) (step 4).

The construct validity, based on known-groups validity, was demonstrated for the knowledge questions, both with and without the clinical case descriptions. On the knowledge questions (Q1–Q17), the average number of correct answers by midwives was significantly higher compared to pediatric nurses (9.7/17 versus 7.1/17, *p* < 0.001). The same pattern was observed when the clinical case descriptions (2.5/5 versus 1.5/5) were added to the analysis (12.2/22 versus 8.6/22, *p* < 0.001). Irrespective of the group allocation, the scores for both midwives and pediatric nurses on the knowledge questions were low.

With regard to the questions on the use of non-narcotic analgesics (acetaminophen, ibuprofen and aspirin), midwives obtained a higher score than the pediatric nurses (5.9/8 versus 4.3/8), while pediatric nurses provided more incorrect answers and more often selected the ‘I don’t know’ option. Pediatric nurses considered the prolonged use of ibuprofen during lactation to be either unsafe (47%) or they did not know the answer (15%). A similar pattern was observed for the use of acetyl salicylic acid for 1–3 days. Related to the questions on narcotic analgesics (tramadol, codeine and oxycodone), a difference between both groups in the accuracy of responses was seen as well. On average, midwives scored better than pediatric nurses (3.8/9 versus 2.7/9, *p* < 0.01). For both groups, there were more uncertainties and errors for questions related to the prolonged use of opioids and for the use of combinations of opioids (tramadol or codeine) with acetaminophen.

The pattern of different total scores observed between both groups for the knowledge questions was also observed for the self-assessment questions. More specifically, a higher but still limited proportion of midwives felt competent providing accurate advice on the use of analgesics during lactation. While this was 52% versus 29% for non-narcotics, the percentages for opioids were lower in both groups but still with a similar pattern (38% versus 20%) (all *p* < 0.01 or stronger). Both groups further reported having received insufficient education on the use of non-narcotic (midwives: 37%; pediatric nurses: 84%) and narcotic analgesics (midwives: 63%; pediatric nurses: 88%) during breastfeeding, with again higher proportions in pediatric nurses. Midwives also assessed their knowledge on non-narcotics and narcotics higher when compared to pediatric nurses (respectively 52% versus 28% and 50% versus 26%; both at least *p* < 0.05), although still being insufficient.

With regard to information sources, midwives answered that they more commonly search information online (79% versus 63%), will verify hospital protocols (87% versus 74%) or will request the advice of a gynecologist (80% versus 53%) or colleague midwife (94%). Midwives also asked less commonly for advice from a pharmacist (37% versus 71%). Compared to midwives, pediatric nurses more often ask for the advice of a pediatrician or neonatologist (97% versus 81%) and less commonly from a colleague pediatric nurse (83% versus 94%). Pediatric nurses more likely requested the advice of a midwife when compared to the reversed (midwife to pediatric nurse) setting (75% versus 21%).

Finally, during the analysis of the pilot data, it turned out that respondents perceived the phrasing of some response categories of some clinical case descriptions as somewhat complex and contradicting (i.e., midwives: case 3, pediatric nurses: case 4). Based on their comments, a slight update of both cases was performed, as highlighted in the instrument provided in the Appendix A.

## 4. Discussion

### 4.1. Main Findings

This study aimed to develop and pilot test an instrument to assess the knowledge of midwives and pediatric nurses in the maternal use of analgesics during lactation. Given the high prevalence of the use of analgesics during lactation and the current lack of a reliable assessment instrument, the development of such a questionnaire was needed, not only to provide insight into the current knowledge of practicing HCPs on this topic, but also to identify educational and healthcare related knowledge gaps and subsequent opportunities to optimize mother–infant health outcomes.

From a methodological perspective, a structured stepwise approach was used to generate an instrument with excellent content and construct validity. Based on the input of HCPs and experts during the focus groups and e-Delphi approach, the total number of questions was reduced from [56 + 4] (midwives) and 7 (pediatric nurses) clinical case descriptions (first version) to [(33 + 2) × 5] clinical case descriptions (final version). In addition, further optimizations with regard to the language used, content, maximal doses or additional options were suggested and subsequently applied in different versions of the instrument. After the e-pilot testing, two case descriptions were perceived by the test public to be insufficiently discrimitive, and we therefore decided to add minor changes to the e-pilot tested version (we refer to the Appendix A, suggested changes highlighted).

After the development and content validation, the instrument was pilot tested by >150 practicing midwives and pediatric nurses, confirming the anticipated substantial differences in knowledge between both groups, with midwives having higher knowledge scores. Midwives also reported feeling more competent to provide advice on the use of analgesics during lactation. Both observations are of course not surprising given midwives’ close(r) relationship with and professional interest in recent mothers and breastfeeding women compared to pediatric nurses who primarily focus on the wellbeing of infants. Despite the observed differences, however, we noted that both groups still have insufficient knowledge, mainly with regard to narcotic analgesics. Given that HCPs with more interest in the use of analgesics during lactation may have been more likely to participate in the pilot study, the knowledge scores could actually be an overestimation of the actual situation in current practice. Hence, the findings point towards a clear need for more attention to this topic in the curricula and postgraduate training of both disciplines.

Finally, during the analysis of the pilot data, it turned out that respondents perceived the phrasing of some response categories of two clinical case descriptions as complex and/or contradicting (i.e., midwives: case 3, pediatric nurses: case 4). Based on their comments, a minor update in the answer options of both cases was provided, as highlighted in the instrument provided in the Appendix A.

### 4.2. Strenghts and Limitations

This study has its strengths. First, the study resulted in the development of a valid instrument to assess the knowledge of, specifically, midwives and pediatric nurses on the use of analgesics during lactation. To the best of our knowledge, this is the first instrument developed on this topic so far. The involvement of practicing midwives and pediatric nurses, as well as experts, during the development phase enhanced the applicability of the instrument in the daily setting, integrating their perspectives, reflections and experiences. Second, the instrument was tested by more than 150 practicing HCPs. The 70% completion rate hereby suggested that HCPs considered the burden related to the completion of the questionnaire acceptable and feasible. The duration taken to complete the questionnaire has been recorded, but was based on the first activity until final submission and was sometimes >10 h or >1 day, as the questionnaire completion could be interrupted.

The study also had some limitations that need to be taken into account. First, the instrument was developed in Belgium, involving only Dutch speaking HCPs and experts and aimed at assessing the knowledge of midwives and pediatric nurses in particular. This limited its generalizability to other settings, types of HCPs and countries where other guidelines exist. Second, additional testing and validation, for example with regard to internal consistency, is needed before the instrument becomes a widely accepted tool for research and educational purposes in the context of analgesics use during lactation. From a methodological perspective, we suggest that further validation should focus on exploring additional possibilities of reliability analyses, such as test–retest approaches, preferably in another cohort with similar characteristics to avoid bias related to recall or memory effects. Third, depending on the applicable guidelines on the use of analgesics during lactation in other countries, the correct answers on the knowledge questions may—slightly—differ across countries. For example, it is well known that therapeutic doses of ibuprofen are associated with an increased risk of bleeding due to platelet mediated effects, but we assessed that this is not of clinical relevance in the setting of lactation related exposure (as this effect is dose dependent, question 2). Hence, prior to the application of the instrument in other countries or settings, attention should be paid to the identification of the correct answers. Finally, respondents of the pilot test were recruited via social media and direct e-mailing. As a result, the risk of selection bias cannot be excluded in this sample and should be taken into account when interpreting the results of the pilot testing.

### 4.3. Future Perspectives

Obviously, the development of an instrument is only a first step, as instruments are intended to be either further developed and validated and subsequently used to guide education and clinical research. Reflections on further development and validation have been provided in the limitations section (see Section 4.2).

From an implementation perspective, the analysis of the pilot results confirmed that there is a difference in knowledge level between midwives and pediatric nurses, with midwives scoring better. However, irrespective of the group allocation, the knowledge scores for both midwives and pediatric nurses were low. As analgesics are commonly used during lactation, it seems that further investment in knowledge diffusion, implementation and access to knowledge is needed. We hereby assume that these needs are not restricted to analgesics, but likely also apply to other groups of commonly used medicines in this setting, including but not limited to antibiotics, antihistamines, sedatives, antidepressants and anti-epileptics [5]. Providing access to information, for example guidance and guidelines on ‘indication driven’ pharmacotherapy such as anesthesia and sedation in breastfeeding women, may support local HCPs in knowledge retrieval and implementation [8,9]. Alternatively, systematic reviews on ‘indication’ driven pharmacotherapy such as the recent one on anti-histamines for allergy can provide HCPs and the public with balanced information on best options or choices in a given setting [23].

However, as has been already mentioned as a limitation, we should realize that there is also still a relevant portion of uncertainties and diversity in guidelines. In a recent assessment of discrepancies with regard to information on the safety of medicines during pregnancy and lactation, Teratology Information Services’ (TIS) recommendations were consistent in only 15/22 of the cases but were more aligned compared to regulatory sources [24]. These inconsistencies also related to analgesics, as ibuprofen was one of the six studied medicines. The authors also concluded that regulatory sources had generally more restrictive recommendations [24]. Unfortunately, a dedicated expertise center or TIS is not yet available in each country, as this is the case for Belgium, inciting HCPs or patients to contact the national Poison Center for specific information [25].

It is therefore recommended that, besides additional validation, further research has to be conducted with this instrument on a larger scale and in various settings to identify the knowledge level and potential differences between HCPs. Such insight is vital to identify discrepancies, to provide targeted training or support, to streamline recommendations and to provide guidance on how to improve the relevant curricula. Related to this last point, it became clear that, at least Dutch speaking, midwives and pediatric nurses in Belgium feel that insufficient education on the use of analgesics during lactation was provided during their training (especially regarding narcotics). This resulted in insufficient knowledge among both groups, again especially related to narcotics. These findings provide a strong call for further curriculum development in this clinical area in the different HCPs disciplines involved in care for lactating women and their infants.

## 5. Conclusions

This paper reports on the development and pilot use of an instrument to assess the knowledge of midwives and pediatric nurses on maternal use of analgesics during lactation and the potential risks for the nursing infant. Therefore, a structured, stepwise approach with the consecutive use of a literature search, focus groups with practicing HCPs, a two-round e-Delphi with experts for item and score content validity and a pilot survey to test construct validity was applied. This multifaceted approach resulted in a valid instrument consisting of 33 questions and five specific clinical case descriptions for both disciplines. In the future, the instrument could be further validated in different settings and used on a large scale for research and educational purposes. As these pilot findings suggest suboptimal knowledge for both professions on this topic, adaptations to their postgraduate curricula and postgraduate training might be warranted.

## Figures and Tables

**Figure 1 ijerph-18-11555-f001:**
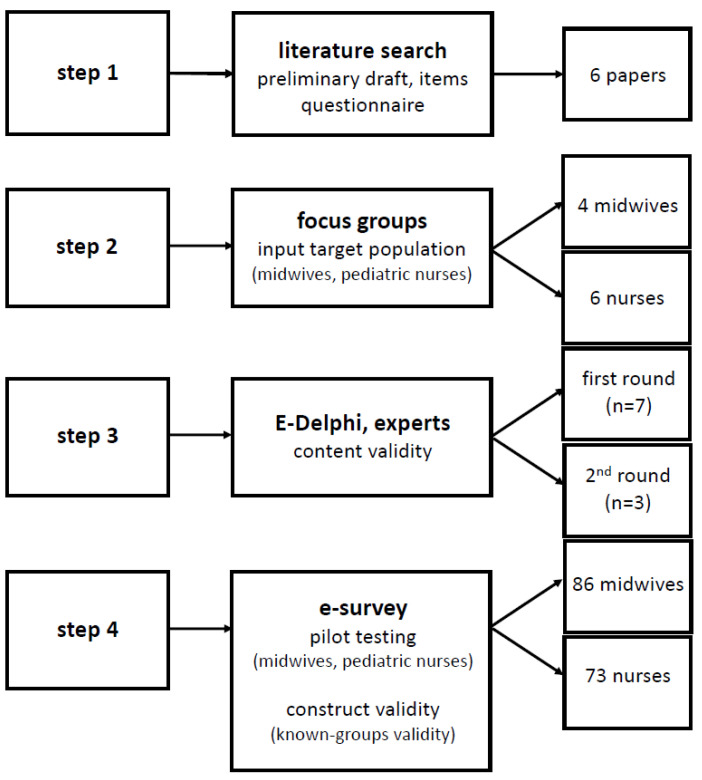
Workflow of the development and testing of the instrument.

**Table 1 ijerph-18-11555-t001:** Characteristics of the six papers retained based on the literature screening, reflecting different item subgroups of knowledge gaps on the maternal use of analgesics during lactation [16,18,19,20,21,22].

Reference	Methods	Main Findings
Al-Sawalha et al. [16]	After a pilot study (*n* = 10), a self-constructed (approach not described) questionnaire was distributed to HCPs (nurses, pharmacists, physicians) in Jordan. This questionnaire had 23 questions (on demographics, attitudes on medicine use during lactation and knowledge) related to the most commonly used medicines during lactation.	904 responses, 44% nurses (no sub-specialties mentioned). 27% advised to always stop or interrupt breastfeeding whenever a lactating mother took any medicine. Awareness on recommendations was lower in nurses (OR 0.21) compared to physicians. 80% of HCPs considered themselves as having a low level of knowledge, even lower in nurses (OR 0.10). A request to add this topic to curricula and professional continuing education.
Spiesser-Robelet et al. [18]	Scoping review on literature sources on breastfeeding mothers’ knowledge, representations, attitudes and behaviors about medicines resulted in 18 papers and 15 studies. Questionnaire development was not assessed, nor discussed.	Most (12/15) studies were quantitative, with HCPs as the target audience and questionnaires were commonly (8/15) used. The studies reflect an almost systematic conflict for the mothers between taking medicines and breastfeeding. Studies describe safety behaviors of breastfeeding women taking medicines, but do not allow them to understand how breastfeeding mothers’ behaviors were constructed. Items were maternal knowledge (*n* = 2), social representations (*n* = 4), attitudes (*n* = 1) and behaviors (incidence, acceptability, or consequence of medicine use during lactation, measures to reduce infant exposure).
Colaceci et al. [19]	State-of-the-art development of a questionnaire, using a mixed methods study, with the construction of the questionnaire based on four categories (experience, medicines versus natural products, access to information and adverse reactions), subsequently administered to 248 pregnant women or mothers.	Women showed three attitudes: discontinue breastfeeding in order to take the medicine, “endure the pain” or use ‘natural products’ as these are perceived to be safer. Information sources for lactation management were pediatricians (46%), midwives (24%) and prescribers (10%), reflecting the relevance of HCPs.
Verstegen et al. [20]	Narrative review, with a focus on the clinical pharmacology of lactation related medicine exposure and methods to assess exposure and effects.	Specific section on the lactation compatibility of analgesics, anesthetics and sedatives. Acetaminophen and non-steroidal, anti-inflammatory drugs are safe. Opioids can be used safely for short-term pain management, with the need for more intense monitoring (lethargy, respiratory depression) when longer treatment duration is needed.
Amundsen et al. [21]	Cross-sectional questionnaire among 401 women with migraine, either pregnant or in postpartum (<18 months). The development of the questionnaire has not been described, but a pilot (*n* = 6) was done, with only minor adaptations afterwards.	The majority severely overestimated the risk associated with migraine medicines during pregnancy or lactation. Women who reported medicine use were more positive and overestimated lesser the risks of such medicines compared with their counterparts.
Wolgast et al. [22]	Questionnaire on the use, perceptions towards the use and perceptions about pregnancy outcomes in association with medicines during pregnancy and lactation. In total, 850 women participated. Its development was based on two questions from a former questionnaire.	The majority (58%) perceived medicines during lactation as (probably) harmful and perceived herbal medicines as less harmful (21%). Women had great confidence in advice form a physician (84%) or midwife (77%).

HCPs: healthcare providers; OR: odds ratio.

## Data Availability

The data as collected have been stored and can be provided based on a reasonable request.

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
