# Peer review of "Development and Pilot Use of a Questionnaire to Assess the Knowledge of Midwives and Pediatric Nurses on Maternal Use of Analgesics during Lactation"

_ijerph, 2021, doi:10.3390/ijerph182111555_

Round 1
Reviewer 1 Report
This paper reports on the step-by-step development of an instrument and pilot testing of the instrument to assess the knowledge of midwives and pediatric nurses on maternal use of analgesics during lactation and the potential risks for the nursing infant.
The steps followed to develop the instrument were clearly stated and presented in a Figure. Based on reviewing the existing literature, focus group discussions with practising midwives and the pediatric nurses and experts input- the instrument was developed. Following instrument development and content validation- the questionnaire was pilot tested, which demonstrates that the authors followed the scientific approaches in constructing and validating an instrument.
The results of the pilot survey with midwives and pediatric nurses demonstrate high construct validity of the instrument, suggesting that the instrument might be used to assess knowledge on lactation-related exposure to analgesics. The pilot findings also suggest that the instrument has potential values in assessing the knowledge of the health care professionals and can be added in the health care professional training as well.
This well-written manuscript can be acce[ted for publication as it is.
Author Response
we are grateful for the very supportive and positive assessment of the paper by the reviewer. There we not comments, so that we have nothing to revise.
Reviewer 2 Report
Minor Revision is needed:
- Page 7, section 3. Results, line 223 - Please use the English term for: „respiratoire distress”
- Page 8, section 3. Results, line 258 - Please correct typographical error of „constuct”
- Page 9, section 3. Results, line 289 - Please indicate the value of „p”: „p ...”
- Page 10, section 4. Discussion, line 320 - Please correct typographical error of „discrimitative”
Author Response
we are very grateful for the very supportive and positive assessment of this paper. we have implemented the comments provided.
- Page 7, section 3. Results, line 223 - Please use the English term for: „respiratoire distress”
Sorry, for the typo, corrected to respiratory
- Page 8, section 3. Results, line 258 - Please correct typographical error of „constuct
Adapted to ‘construct’
- Page 9, section 3. Results, line 289 - Please indicate the value of „p”: „p ...”
Added for both outcome variables
- Page 10, section 4. Discussion, line 320 - Please correct typographical error of „discrimitative”
Adapted to discrimitive